# Compensate, Don't Reconstruct: Parameter- and Data-Efficient 2-bit LLM Quantization

## Abstract

The substantial memory footprint of large language models (LLMs) remains a key barrier to their on-device deployment. 2-bit quantization is a promising solution; however, current methods impose a difficult trade-off between the high accuracy of training-intensive Quantization-Aware Training (QAT) and the efficiency of lower-performing Quantization Error Compensation (QEC). Our analysis of QEC reveals a critical insight: its effectiveness is more dependent on minimizing activation discrepancy than weight discrepancy alone. Building on this, we introduce LG-QEC, a framework that significantly enhances the compensation process. LG-QEC combines a hybrid adapter and a local-global optimization strategy to directly align activations and suppress quantization errors. Experiments show LG-QEC achieves accuracy comparable to state-of-the-art QAT methods while using only a fraction of the training token budget and trainable parameters. This work successfully bridges the gap between efficiency and performance, enabling accurate and practical 2-bit LLMs.

## 1 Introduction

The remarkable proliferation of Large Language Models (LLMs) has unlocked unprecedented capabilities across numerous domains Kamalloo et al. (2023); Rozière et al. (2024); Zhang et al. (2024). However, this advancement comes at the cost of immense model size, with leading models now comprising hundreds of billions of parameters OpenAI (2023). This scale poses a significant barrier to their deployment in resource-constrained environments, such as mobile devices, where on-device inference is crucial for achieving latency, privacy, and offline functionality Gunter et al. (2024); Li et al. (2025). To bridge this gap, quantization has become an indispensable technique for model compression. While 8-bit and 4-bit quantization offer substantial memory savings, ultra-low-bit quantization, particularly at the 2-bit level, represents the frontier for maximizing efficiency and enabling complex models to operate within the tight memory budgets of edge hardware Liu et al. (2025); Chen et al. (2025); Lee et al. (2025b).

The path to effective 2-bit quantization is fraught with challenges. Post-Training Quantization (PTQ) methods, even advanced approaches like QuIP# Tseng et al. (2024), often result in an unacceptable degradation of model accuracy. Consequently, Quantization-Aware Training (QAT) has been the dominant strategy, with methods like ParetoQ Liu et al. (2025) achieving performance comparable to that of 4-bit quantization and approaching the level of full-precision models. However, this accuracy recovery requires drastically updating all weights to overcome the significant quantization error—a process of 'weight reconstruction'. This necessitates substantial computational resources, including a prohibitive memory footprint and tens of billions of training tokens. While techniques like EfficientQAT Chen et al. (2025) have been proposed to reduce memory usage and long training times, they fail to match the performance of QAT trained on a 30B token budget.

As an alternative, Quantization Error Compensation (QEC) utilizes lightweight, trainable adapters to directly 'compensate' for weight quantization error. This approach has demonstrated a greater accuracy recovery effect on an equally low training budget. However, a fundamental trade-off exists: reducing the number of trainable parameters to minimize adapter overhead often leads to a decline in accuracy recovery performance.

This paper aims to resolve this trade-off and significantly improve QEC's accuracy recovery. We present a systematic study analyzing the impact of 1) adapter structure, 2) training token budget, and 3) quantizer design. Through extensive experiments, we identify three key findings:

- **Finding 1:** An adapter's performance is critically linked to its structure and initialization; a hybrid design combining weight- and error-based signals is superior because it minimizes activation discrepancy, which is a better predictor of final accuracy than weight discrepancy alone (Sec. 4.2).
- **Finding 2:** While scaling up either the adapter budget or training token budget improves accuracy, increasing the training token budget forces an inefficient adaptation where the model reduces activation discrepancy at the cost of sacrificing weight fidelity (Sec. 4.3).
- **Finding 3:** The choice of quantizer fundamentally shapes the adaptation process, as a refined vector quantizer suppresses initial weight discrepancy, enabling a stable, compensation-driven regime that allows adapters to efficiently correct activation errors (Sec. 4.4).

These findings establish a core principle: effective 2-bit QEC depends on first *stabilizing weights* to create a foundation for precise *activation alignment*. We materialize this principle in LG-QEC, a novel framework that systematically orchestrates quantizer design, adapter architecture, and a local-global optimization strategy toward a single goal: minimizing activation discrepancy. By first using a vector quantizer to suppress initial weight errors and then employing a two-stage process that decouples local activation alignment from global fine-tuning, our approach achieves a perplexity of 8.1 on WikiText-2 Merity et al. (2016) with only 16M training tokens on the 2-bit Llama-3-8B model. This result matches the state-of-the-art performance of QAT trained with 30B tokens Liu et al. (2025). Moreover, LG-QEC improves Commonsense Question Answering (CSQA) Talmor et al. (2019) accuracy by 0.65% over training without local optimization and matches or surpasses QEC trained with 64M tokens on both CSQA and MMLU Hendrycks et al., demonstrating its effectiveness under extremely data-efficient settings.

## 2 RELATED WORK

The immense size of Large Language Models (LLMs) necessitates the development of effective compression strategies for their deployment on resource-constrained devices. The landscape of low-bit quantization is broadly divided into two main paradigms: Post-Training Quantization (PTQ), which quantizes a pre-trained model without retraining, and Quantization-Aware Training (QAT), which incorporates the quantization process into the fine-tuning stage to mitigate accuracy degradation. While PTQ methods have shown progress, achieving high accuracy at extreme bit widths, such as 2-bit, often requires more sophisticated training-based approaches.

**Quantization-Aware Training (QAT) .** QAT is the most prevalent method for accuracy recovery in ultra-low-bit weight quantization. Its core strategy involves simulating quantization effects during fine-tuning, allowing all weight parameters to be updated to compensate for potential accuracy loss. Recent advancements have focused on improving its effectiveness and efficiency. For instance, ParetoQ provides a unified framework for analyzing low-bit quantization, revealing a critical learning transition between 2-bit and 3-bit precision. UPQ Lee et al. (2025b) addresses the practical challenge of data access by combining block-wise PTQ with a distillation-based QAT. Furthermore, EfficientQAT Chen et al. (2025) tackles the computational overhead of traditional QAT by proposing a two-phase algorithm that significantly reduces training cost.

**Quantization Error Compensation (QEC).** QEC has emerged as a popular, parameter-efficient alternative to recover accuracy loss from quantization error . These methods typically freeze the quantized base model and train lightweight adapters to correct errors. They can be characterized by their initialization strategies and adapter structures.

- *Initialization Strategy:* Two main approaches exist. The Error-initialization strategy, used by LoftQ and LQ-LoRA Guo et al., initializes an adapter to approximate the quantization error matrix $(W - Q)$ explicitly. In contrast, the Weight-init strategy, employed by PiSSA, preserves the most salient components of the original weights by initializing the adapter with their principal singular values and vectors.

- *Adapter Structure:* QEC methods leverage different adapter forms. Low-rank adapters (LoRA) are a common choice for capturing distributed, global error patterns. As a more parameter-efficient alternative, sparse adapters focus on updating a minimal subset of parameters to correct localized, critical errors. Methods like RoSA Nikdan et al. have explored a hybrid approach, jointly training a low-rank and a sparse component to capture both global and fine-grained updates.

## 3 MOTIVATION

While QAT and QEC both aim to reconcile low-precision weights with high accuracy, they represent fundamentally different philosophies in addressing quantization error, especially at the 2-bit level. This section compares these approaches through the lens of 'reconstruction' vs. 'compensation'.

**The Challenge of Full QAT as 'Reconstruction'.** QAT is recognized as a powerful method for recovering accuracy in ultra-low-bit settings. Leading methods, such as ParetoQ, demonstrate that 2-bit models can achieve high performance. However, this effectiveness comes at a significant cost. As shown in Fig. 1(a), QAT requires maintaining a full-precision (FP) master copy of all weights during training, leading to a prohibitive memory footprint, and often necessitates tens of billions of training tokens for optimal results (Fig. 1(b)). The root of this inefficiency lies in the 'reconstruction' process that 2-bit quantization forces upon the model. As identified by ParetoQ, the quantization error is so significant that the model's original weight distribution is effectively destroyed. Consequently, the training process is not merely fine-tuning but a substantial undertaking to relearn new functional representations from scratch by making significant changes to the weight parameters. This reconstruction is highly data-dependent; with a limited budget of 1M fine-tuning tokens, QAT fails to complete this process, resulting in an impractically high perplexity (PPL) of over 1100 (Fig. 1(c)).

**The Efficiency of QEC as 'Compensation'.** In contrast, QEC offers a more efficient paradigm by reframing the problem as 'compensation'. Instead of reconstructing the entire model, QEC freezes the low-bit weight backbone and uses lightweight, trainable adapters to compensate for the quantization error directly. This approach dramatically reduces the trainable parameter count, leading to a $17.7\times$ reduction in training memory. Because it performs a targeted correction rather than a complete reconstruction, QEC is significantly less reliant on extensive training token budget (Fig. 1(a-b)). As shown in Fig. 1(c), QEC achieves an intense PPL of approximately 14 with the same 1M training tokens that left QAT ineffective. This demonstrates the clear advantage of a compensation-based strategy in resource-constrained scenarios. While methods like EfficientQAT Chen et al. (2025) attempt to find a middle ground by applying QAT locally, they do not entirely escape the overhead of reconstruction and still fall short of large-scale QAT's peak performance.

This study is motivated by the unrealized potential of a proper compensation-driven approach. We build upon the inherent efficiency of QEC and propose to significantly enhance its accuracy recovery capabilities through a systematic investigation into three key areas: 1) adapter construction, 2) quantizer optimization, and 3) local-global optimization.

## 4 OBSERVATIONS

In this section, we systematically analyze the factors that govern the effectiveness of QEC. We begin by formalizing different adapter structures and initialization strategies (Sec. 4.1) and examine how they influence QEC (Sec. 4.2). To gain deeper insight into why performance differences arise, we complement accuracy metrics with discrepancy analysis in both weight and activation space, measured relative to the FP baseline. We then investigate how scaling the trainable parameter budget and the number of the training tokens affects accuracy recovery (Sec. 4.3), and finally evaluate how the choice of quantizer fundamentally constrains the attainable performance (Sec. 4.4).

### 4.1 ADAPTER SETTINGS

We systematically investigate how different adapter initialization methods and structural choices affect QEC. Unlike prior studies that primarily restrict their analysis to either low-rank or sparse adapters Li et al. (a); Guo et al.; Zhang et al., our study explicitly incorporates the hybrid design,

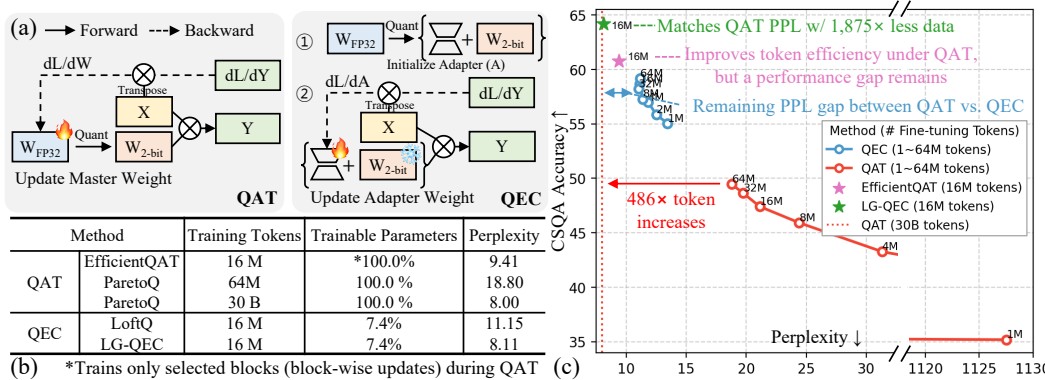

Figure 1: (a) Comparison of forward/backward pass in QAT and QEC. (b) Comparison of training token usage, trainable parameter ratio, and Wikitext-2 perpleixty (PPL) across QAT and QEC methods. (c) Perplexity and CSQA accuracy of QEC and QAT across training token budgets.

---

**Algorithm 1** Adapter Initialization under Quantization with Parameter Budget

---

**Require:** Full-precision weight $W \in \mathbb{R}^{m \times n}$, quantizer $\text{Quant}(\cdot)$, parameter budget $p\%$ of $|W|$
**Ensure:** Approximate decomposition $W \approx Q + A$

1: **Notation:**

| | | | | |
|---|---|---|---|---|
| $Q$ | quantized base weight | | $A$ | adapter (L, S, or LS) |
| Top-$k(X)$ | $k$ largest-magnitude elements of $X$ | | $\text{SVD}_r(X)$ | rank-$r$ truncated SVD of $X$ |
| L | low-rank adapter from $\text{SVD}_r(\cdot)$ | | | $(U_r S_r V_r^\top)$ |
| LS | hybrid adapter (L+S) | | S | sparse adapter from Top-$k(\cdot)$ |

2: **Budget allocation:**
3: $k \leftarrow \lfloor p\% \times (m \cdot n) \rfloor$      // number of sparse elements
4: $r \leftarrow \lfloor k/(m+n) \rfloor$      // LoRA rank (approx. same params)
5: For LS, allocate $r/2$ and $k/2$ to each branch *to maintain the total parameter budget $p\%$*.
6: **Case 1: Zero-Init**
7: $A \leftarrow \begin{cases} \mathcal{N}(0, \sigma^2)^{m \times r} \cdot 0^{r \times n} & \text{(L), standard LoRA zero initialization} \\ 0^{m \times n} & \text{(S), } k \text{ sparse positions initialized to zero} \\ \mathcal{N}(0, \sigma^2)^{m \times (r/2)} \cdot 0^{(r/2) \times n} + 0^{m \times n} & \text{(LS), zero-initialized low-rank + sparse adapter} \end{cases}$
8: **Case 2: Error-Init**
9: $Q \leftarrow \text{Quant}(W)$
10: $E \leftarrow W - Q$
11: $A \leftarrow \begin{cases} \text{SVD}_r(E) & \text{(L), low-rank approximation of quantization error} \\ \text{Top-}k(E) & \text{(S), sparse selection from quantization error} \\ \text{SVD}_{r/2}(E) + \text{Top-}k/2(E) & \text{(LS), low-rank + sparse adapter initialized from error} \end{cases}$
12: **Case 3: Weight-Init**
13: $A \leftarrow \begin{cases} \text{SVD}_r(W) & \text{(L), low-rank approximation of full-precision weight} \\ \text{Top-}k(W) & \text{(S), sparse selection from full-precision weight} \\ \text{SVD}_{r/2}(W) + \text{Top-}k/2(W) & \text{(LS), low-rank + sparse adapter initialized from weight} \end{cases}$
14: $Q \leftarrow \text{Quant}(W - A)$

---

where both low-rank and sparse adapters are jointly considered within the same parameter budget. In addition, all comparisons are conducted under a fixed parameter budget to ensure that the evaluation across adapter types and initialization strategies remains fair and controlled.

**Adapter initialization methods.** We consider a design space comprising three initialization strategies and three adapter structures, as summarized in Algorithm 1. The initialization strategies include *Zero-Init*, where all adapter parameters are initialized to zero regardless of structure; *Error-Init*, which uses the quantization error $W - \text{Quant}(W)$ to initialize the adapter via either a low-rank SVD approximation or sparse top-$k$ selection; and *Weight-Init*, which directly derives adapter weights from the FP weights $W$, quantizing only the residual $W - A$.

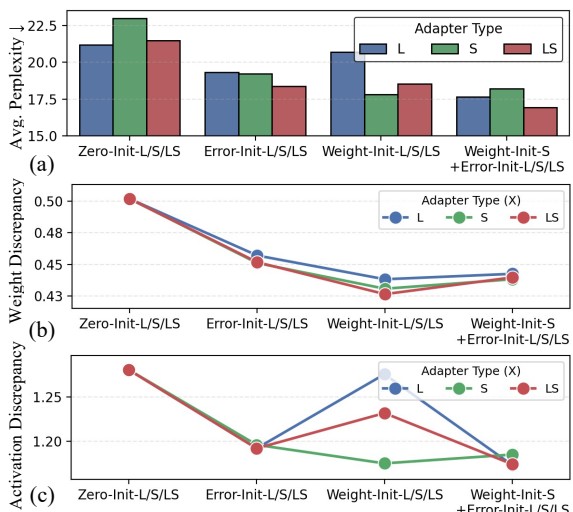

Figure 2: Example procedure for using Weight-Init and Error-Init together (Weight-Init-S + Error-Init-LS). The process constructs a sparse adapter, quantizes the residual, and compensates remaining errors with both a low-rank adapter and a second sparse adapter. The final result combines the frozen quantized weight and three adapters while keeping the total parameter budget fixed.

**Adapter types.** The adapter structures under evaluation are: (1) *L (low-rank)*, employing a rank-$r$ LoRA-style structure; (2) *S (sparse)*, which selects $k$ non-zero elements based on magnitude; and (3) *LS (hybrid)*, which combines both low-rank and sparse branches within a fixed parameter budget. In the hybrid case, the total budget is split evenly, assigning $r/2$ and $k/2$ to the low-rank and sparse components respectively.

To clearly represent different adapter settings, we adopt a unified notation that concatenates the initialization method and the adapter type. For example, **Error-Init-S** denotes a sparse adapter initialized from quantization errors, and **Weight-Init-LS** represents a hybrid adapter where both low-rank and sparse components are constructed from FP weights. Visualizations of adapters, base weights, and residual errors under different initialization methods are provided in Appendix A.2.

An important aspect of our design space is that initialization methods are not mutually exclusive. Weight-based and error-based initialization can also be applied in combination, and the unified notation naturally extends to such settings. For example, **Weight-Init-S + Error-Init-LS** denotes a sparse adapter initialized from FP weights and additional hybrid adapters initialized from residual errors. Fig. 2 illustrates this combined case under a fixed parameter budget.

## 4.2 IMPACT OF ADAPTER INITIALIZATION METHOD AND STRUCTURE

We next examine how adapter initialization strategies and structural design choices influence QEC performance. Our key finding is that reducing weight discrepancy alone is insufficient; minimizing activation discrepancy provides a more reliable indicator of final performance. We support this with two complementary discrepancy metrics that quantify deviations from the FP model:

$$D_{\text{weight}} = \frac{\|W - (Q + A)\|_F}{\|W\|_F}, \quad (1)$$

$$D_{\text{activation}} = \frac{\|X - X_q\|_F}{\|X\|_F}, \quad (2)$$

where $W$ and $Q + A$ denote the FP weight and the quantized weight with adapter compensation, respectively. $X$ is the input activation from FP inference, and $X_q$ is the corresponding activation from the quantized model. $D_{\text{weight}}$ captures parameter-level deviation, while $D_{\text{activation}}$ reflects functional misalignment. Unless otherwise specified, both metrics are averaged across all Transformer layers. We evaluate these effects through short QEC fine-tuning on 256 samples from C4 Raffel et al. (2019) with sequence length 512. Fig. 3 reports the results: (a) PPL after fine-tuning, (b) $D_{\text{weight}}$, and (c) $D_{\text{activation}}$ measured at initialization.

Figure 3: (a) Perplexity after fine-tuning with different adapter initialization methods. (b) Weight and (c) activation discrepancy after adapter initialization.

**Model accuracy impact.** As shown in Fig. 3(a), providing informative initialization—either from FP weights (Weight-Init) or from quantization errors (Error-Init)—substantially improves PPL compared to Zero-Init, which lacks any prior signal. Among individual methods, Weight-Init-S offers strong accuracy improvements, while their combination—Weight-Init-S + Error-Init-LS—achieves the lowest PPL overall. This result indicates that combining weight-based and error-based signals, even under the same parameter budget, yields superior performance. Also, the hybrid adapter structure (LS) can offer improved performance over purely low-rank or sparse structures. This suggests that low-rank and sparse branches may capture complementary aspects of quantization error, and that their joint use, when properly initialized, can potentially offer a more robust mechanism for error mitigation.

**Discrepancy analysis.** Fig. 3(b) shows that both Weight-Init and Error-Init effectively reduce $D_{\text{weight}}$ compared to Zero-Init, with Weight-Init-S achieving the lowest discrepancy. However, tighter alignment in weight space does not always translate to improved downstream performance. For instance, the combined configuration (Weight-Init-S + Error-Init-LS) incurs a slightly higher $D_{\text{weight}}$ than Weight-Init-S alone, yet yields better PPL. This apparent discrepancy is explained by differences in activation behavior, as shown in Fig. 3(c). Within Weight-Init, sparse adapters more effectively reduce $D_{\text{activation}}$ than their low-rank counterparts. Although LoRA can closely match the original weights, they often induce functional misalignment—underscoring a key limitation: accurate weight reconstruction does not necessarily preserve activation semantics. The combined approach mitigates this limitation by distributing the parameter budget across complementary components: the sparse branch initialized from weights minimizes $D_{\text{weight}}$, while the error-based low-rank and sparse branches further reduce $D_{\text{activation}}$.

## 4.3 Impact of Trainable Parameter and Training Token Budget

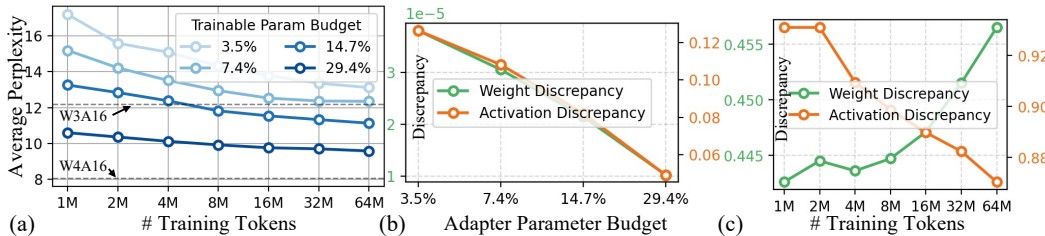

(a)        # Training Tokens     (b)    Adapter Parameter Budget    (c)    # Training Tokens

Figure 4: Impact of adapter budget and number of training tokens on QEC performance. (a) Perplexity after full QEC training with varying numbers of training tokens. (b) Average weight and activation discrepancies across adapter parameter budgets with a 64M training tokens. (c) Average weight and activation discrepancies across training tokens under a 3.5% adapter parameter budget.

We now investigate how scaling the trainable parameter budget and the amount of training tokens influences QEC performance. All experiments in this section begin from the best-performing adapter configuration identified in Sec. 4.2—Weight-Init-S + Error-Init-LS—which already provides a strong starting point by combining structural diversity with informative initialization.

**Adapter size.** As shown in Fig. 4(a), larger adapters yield lower PPL for a fixed number of training tokens, indicating improved adaptation capacity. Consistently, Fig. 4(b) shows that both $D_{\text{weight}}$ and $D_{\text{activation}}$ decrease as the adapter budget increases. This suggests that increasing capacity allows QEC to better approximate both parameter- and function-level behaviors. However, this improvement comes at the cost of increased memory usage during both training and inference, limiting its practicality under strict resource constraints.

**Training tokens.** Scaling the number of training tokens also leads to consistent gains in QEC performance. As shown in Fig. 4(a), with a fixed adapter budget of 7.4%, increasing the training tokens up to 32M is sufficient to reach the perplexity level of W3A16. However, as depicted in Fig. 4(c), this improvement does not arise from better alignment in weight space. In fact, $D_{\text{weight}}$ slightly increases as training proceeds with more tokens. Instead, $D_{\text{activation}}$ steadily decreases, closely tracking the improvements in PPL. These results expose a key limitation of current QEC training dynamics. While high-quality initialization reduces both $D_{\text{weight}}$ and $D_{\text{activation}}$ (Sec. 4.2), we observe that QEC continues to lower $D_{\text{activation}}$ during training at the expense of increasing $D_{\text{weight}}$. This trend implies

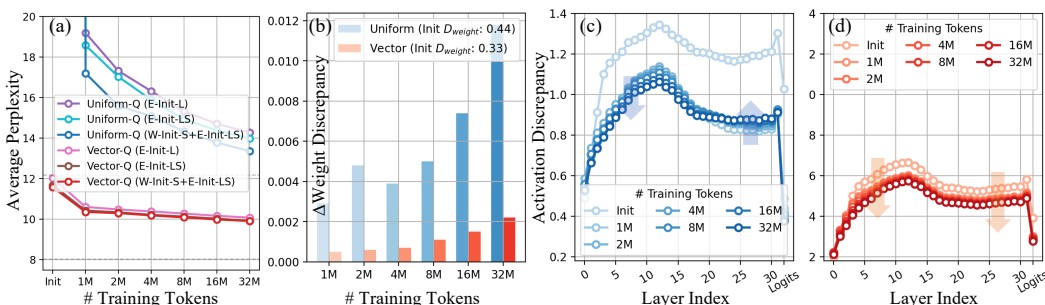

Figure 5: Impact of quantizer choice on QEC. (a) PPL across adapter types and training tokens. (b) Change in weight discrepancy from adapter initialization during training token scaling. (c–d) Activation discrepancy under uniform and vector quantizers as the number of training tokens increases (Adapter type: Weight-Init-S+Error-Init-LS).

that the model sacrifices weight fidelity in pursuit of activation alignment, ultimately undermining the stability of adaptation—a behavior that counters the intended role of initialization as a foundation for robust performance recovery.

## 4.4 IMPACT OF QUANTIZER DESIGN

We next investigate how quantizer design shapes the adaptation dynamics of QEC, particularly in terms of weight stabilization and activation discrepancy minimization. To this end, we compare two representative 2-bit quantizers applied under identical adapter configurations: (i) an asymmetric uniform quantizer with group size 64, and (ii) a vector quantizer based on incoherence processing with codebooks Tseng et al. (2024). Although both operate at the same bit precision, their effects on weight behavior, activation alignment, and final PPL diverge significantly.

Fig. 5(a) highlights a striking difference in PPL trajectories. Vector quantization achieves substantially lower PPL already at initialization, with rapid convergence within the first 1M–2M tokens. This early stabilization implies that most of the functional adaptation occurs before any extensive weight changes are necessary. In contrast, uniform quantization starts from a much higher PPL and improves only gradually.

Discrepancy analyses in Fig. 5(b–d) support this distinction. Uniform quantization leads to large $D_{\text{weight}}$ values, indicating significant weight modification throughout training. Although this helps reduce activation discrepancies over time, it comes at the cost of poor weight-level stability. Furthermore, Fig. 5(c) shows that activation alignment under uniform quantization often involves complex cross-layer trade-offs, with some deeper layers exhibiting drift despite overall improvements in output similarity. These trends suggest that under uniform quantization, QEC attempts to reduce activation discrepancy, but does so at the cost of disrupting weight-level stability—leading to unstable adaptation dynamics. In contrast, vector quantization preserves the original weight structure more faithfully, resulting in smaller $D_{\text{weight}}$ across training tokens. This inherent stability allows QEC to operate in a compensation-oriented regime, where adapters make localized, fine-grained adjustments to reduce residual activation discrepancies. Indeed, Fig. 5(d) shows consistently low $D_{\text{activation}}$ throughout training, without the layer-level drift seen in the uniform case.

The effectiveness of hybrid initialization (Weight-Init-S + Error-Init-LS) also varies across quantizers. Under uniform quantization, hybrid initialization yields substantial performance gains, reflecting the need to preemptively reduce weight discrepancy. However, under vector quantization—where weight errors are already suppressed—such hybridization offers only marginal benefit, highlighting how quantizer quality modulates the role of initialization.

Taken together, these results show that quantizer design fundamentally shapes the QEC adaptation regime. Uniform quantization necessitates weight reconstruction and induces unstable cross-layers alignment. Vector quantization, by contrast, enables a more desirable pathway: it stabilizes weights early and allows adapters to focus on minimizing activation discrepancies. This finding reinforces the importance of targeting activation alignment once weight-level errors are constrained, motivating our subsequent approach that incorporates local optimization to enhance QEC performance.

## 5 Employing Local Optimization to Boost QEC Performance

The analyses in Sec. 4 reveal a consistent trend: although QEC benefits from strong initializations and increased training tokens, its performance ultimately depends on how effectively it reduces activation discrepancies during training. Crucially, when weight-level errors are large—as is often the case under uniform quantization—QEC struggles to stabilize, frequently altering weights in a manner that undermines consistent adaptation. Conversely, employing refined quantizers such as vector quantization suppresses weight discrepancies from the outset, enabling QEC to concentrate on correcting activation-level mismatches.

These findings suggest a division of roles in effective QEC: quantizer design should minimize weight-level discrepancies, while adapters should focus on locally reducing activation mismatches without disrupting weight stability. However, standard end-to-end QEC fine-tuning forces the model to solve both problems simultaneously. As observed in Sec. 4.3, this joint optimization can lead to unstable dynamics where the model sacrifices weight fidelity to improve activation alignment. Based on this insight, we introduce **LG-QEC**, a two-stage optimization procedure that explicitly *decouples*

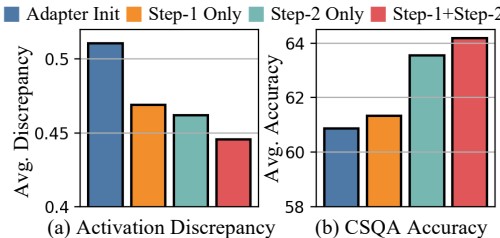

Figure 6: Effectiveness of local optimization. Step-1 reduces $D_{activation}$ comparably to Step-2 alone, while Step-1+Step-2 achieves both the lowest discrepancy and the highest CSQA accuracy.

these competing objectives. By separating the initial, coarse-grained alignment of activations from the subsequent refinement of residual errors, **LG-QEC** achieves a more stable and efficient compensation process. All experiments are conducted with vector quantization and Error-Init-LS adapters, ensuring that weight errors are already minimized at initialization.

- **Step-1 (Local optimization):** A brief calibration phase with 4M tokens and half the total parameter budget focuses on aligning activations between the quantized and FP models. This local adaptation reduces activation discrepancy with minimal weight changes, providing a stable starting point for training.
- **Step-2 (Global optimization):** Standard QEC fine-tuning is then applied with 16M tokens and the full parameter budget. Building upon the activation-aligned model from Step-1, this stage refines residual errors more effectively under the same budget constraint.

**Effectiveness.** Fig. 6 summarizes the results of our two-stage procedure, evaluated under a fixed 16M token budget. We compare our LG-QEC approach—a 4M token local optimization phase (Step-1) followed by a 12M token fine-tuning phase (Step-2)—against a baseline ('Step-2 Only') that uses the entire 16M tokens for standard fine-tuning. The combined LG-QEC method yields both the lowest activation discrepancy and the highest CSQA accuracy, outperforming either stage in isolation. While Step-1 alone effectively reduces discrepancy, it provides insufficient accuracy, demonstrating the strength of the combined approach. This result validates our hypothesis: once weight discrepancy is controlled, explicitly targeting activation alignment enables more efficient and stable QEC adaptation.

## 6 Experiments

We evaluate the effectiveness of refined quantizers and local optimization as preprocessing strategies for QEC on the Llama-3-8B model Meta (2024). Experimental setup, including training configurations and benchmark details, is provided in Appendix A.1.

### 6.1 Quantization Error Compensation Results

Table 1 demonstrates that both quantizer choice and the presence of local optimization critically affect downstream performance. Uniform quantization yields consistently poor results, with PPL remaining high (>11 on Wikitext-2) and CSQA accuracy plateauing below 60% even as the training token increases from 16M to 64M. In contrast, vector quantization substantially improves perfor-

| Quantizer | Local Optim. | # Fine-tuning Trained Tokens | PPL ↓ | | CSQA Accuracy ↑ | | | | | | MMLU↑ |
|---|---|---|---|---|---|---|---|---|---|---|---|
| | | | Wiki2 | C4 | ARC-C | ARC-E | Hellaswag | PIQA | Winorande | Avg. | Avg. |
| Baseline | | | 6.14 | 8.88 | 50.43 | 80.22 | 60.15 | 79.54 | 72.93 | 68.65 | 65.13 |
| Uniform | – | 16M | 11.15 | 13.92 | 34.64 | 68.14 | 49.86 | 74.48 | 66.30 | 58.68 | 40.17 |
| | – | 32M | 11.04 | 13.69 | 35.07 | 68.90 | 49.88 | 73.83 | 63.46 | 58.23 | 43.18 |
| | – | 64M | 11.16 | 13.53 | 35.67 | 68.64 | 50.76 | 75.41 | 65.35 | 59.17 | 41.23 |
| Vector | – | 16M | 8.15 | 11.26 | 42.66 | 74.37 | 55.08 | 76.55 | 69.06 | 63.54 | 55.06 |
| | – | 32M | _8.11_ | _11.15_ | 41.98 | 74.79 | 55.45 | 76.66 | 69.06 | 63.59 | 55.37 |
| | – | 64M | **8.08** | **11.04** | **43.43** | _75.59_ | **55.72** | _76.99_ | **69.53** | **64.25** | _55.63_ |
| | ✓ | **16M** | _8.11_ | 11.19 | _43.09_ | **75.63** | _55.52_ | **77.20** | **69.53** | _64.19_ | **55.65** |

Table 1: Impact of local optimization under training token scaling.

mance across all training token sizes, reducing PPL by more than 2 points on both Wikitext-2 and C4 while delivering ∼4% higher CSQA accuracy than uniform quantization.

Crucially, local optimization yields substantial gains even with a small training token budget: with only 16M tokens, it achieves perplexity comparable to 64M-token training and raises CSQA accuracy to 64.19%, matching or exceeding the best results without local optimization. This advantage also appears on the 5-shot MMLU benchmark, where 16M tokens with local optimization outperform 64M tokens without it, underscoring that activation alignment is more critical than merely increasing the number of training tokens.

These results confirm that vector quantization establishes a stable foundation by minimizing weight discrepancies, while local optimization efficiently mitigates activation mismatches. Together, they enable QEC fine-tuning to achieve state-of-the-art PPL and accuracy under significantly reduced training token budgets.

| Method | | # Fine-tuning Trained Tokens | Model Size(GB) | PPL↓ Wiki2 |
|---|---|---|---|---|
| Baseline | | | 16 | 6.1 |
| PTQ | RTN | - | | 2.2E+4 |
| | OmniQuant | - | | 61.8 |
| | QuIP# | - | | 12.7 |
| | QuaRot | - | | 15.0 |
| | GPTQ | - | 2.8 | 2.1E+2 |
| | AWQ | - | | 1.7E+6 |
| | Slim-LLM | - | | 39.7 |
| | DB-LLM | - | | 13.6 |
| | PB-LLM | - | | 24.7 |
| QAT | ParetoQ | 30B | 2.8 | 8.0 |
| | EfficientQAT | 16M | | 9.4 |
| QEC | RILQ(LoftQ) | 0.4M | 3.4 | 18.0 |
| | RILQ(LoftQ) | 32M | 3.4 | 13.2 |
| | LG-QEC | 16M | 3.8 | 8.1 |

Table 2: PPL of Llama-3-8B under 2-bit quantization using PTQ, QAT, and QEC.

## 6.2 2-BIT COMPARISON WITH PTQ AND QAT

Table 2 summarizes the 2-bit quantization results on Llama-3-8B. PTQ methods, while requiring no training tokens, suffer from severe degradation at 2-bit precision: PPL exceeds $10^4$ in most cases, rendering them impractical for language modeling. QAT alleviates this issue by jointly updating all weights during training, reducing PPL below 10. For example, ParetoQ achieves a PPL of 8.0, but only by consuming 30B training tokens, which entails prohibitive computational cost. Meanwhile, QECs such as RILQ Lee et al. (2025a) reduce PPL by optimizing small, full-precision adapters, yet scale poorly with training token budget, even at 32M tokens. In contrast, our proposed approach, **LG-QEC**, which combines vector quantization with hybrid adapter structrue and local optimization, achieves a PPL of 8.1 on Wikitext-2 and 11.2 on C4 with only 16M training tokens, outperforming previous QEC methods.

## 7 CONCLUSION

This paper identifies and validates a core principle for 2-bit LLM quantization: the most efficient path to recovering performance is not to reconstruct weights, but to compensate for errors by directly aligning activation distributions. Our systematic analysis reveals that while suppressing weight discrepancy is necessary for stable adaptation, minimizing activation discrepancy is the ultimate driver of model accuracy. This insight motivates our proposed framework, **LG-QEC**, which combines refined quantizers with local activation alignment to exploit this principle. Experiments show that **LG-QEC** achieves superior 2-bit performance with far fewer parameters and training token budget than PTQ and QAT baselines.

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

Figure 7: Visualization of Zero-Init and Weight-Init adapters. For a base weight $W \in \mathbb{R}^{64 \times 64}$ with a 6% parameter budget, allocated as follows; 6% sparse parameters for Weight-Init-S, rank-2 for Weight-Init-L, and 3% sparse praramter plus rank-1 for Weight-Init-LS.

# A APPENDIX

## A.1 EXPERIMENTAL DETAILS

**Benchmarks.** We report perplexity on WikiText-2 Merity et al. (2016) and C4 Raffel et al. (2019), and accuracy on five commonsense question answering benchmarks: ARC-Challenge Clark et al. (2018), ARC-Easy Clark et al. (2018), HellaSwag Zellers et al. (2019), PIQA Bisk et al. (2019), and WinoGrande Sakaguchi et al. (2019), as well as 5-shot accuracy on MMLU Hendrycks et al.. Perplexity is measured with a sequence length of 2048 tokens. In Table 2, the results for OmniQuant, QuIP#, QuaRot, and RILQ are obtained from their publicly available codebases, while the other results are taken from the respective papers and Huang et al. (2024).

**Training settings.** Both local optimization and QEC fine-tuning are performed on C4 with a maximum token length of 2048. Local optimization is conducted with approximately 4M training tokens in a block-wise manner which enables efficient implementation, similar to previous PTQ methods Frantar et al. (2023); Li et al. (b). For QEC fine-tuning, we allocate adapters corresponding to 7.4% of the total model parameters, ensuring a consistent parameter budget across all experiments. For both stages, we sweep the learning rate from $1 \times 10^{-5}$ to $3 \times 10^{-4}$ under a cosine schedule.

## A.2 VISUALIZATION OF WEIGHT-INIT ADAPTER

We visualize the behavior of Weight-Init adapters using a matrix $W \in \mathbb{R}^{64 \times 64}$. The weights are quantized with a uniform quantizer using a group size of 64 (i.e., row-wise grouping). With Zero-Init, the adapter $A$ is initialized to zero (a), so $W - A$ is identical to the original $W$ (e). Using SVD, the Weight-Init-L isolates the outlier row in $W$ (b), yielding a residual $W - A$ without the outlier (f). Because outliers induce large weight discrepancy after quantization (i), their removal with Weight-Init-L reduces the corresponding row's discrepancies (j), although other rows may still exhibit nontrivial gap. In contrast, Weight-Init-S extracts high magnitude parameters for each group of $W$ (c), shrinking the dynamic range within each group (g) and thereby lowering the overall weight discrepancy (k); however, it does not fully remove an entire outlier row as effectively Weight-Init-L. Combining the two (Weight-Init-LS) captures the outlier structure while also reducing the per-group

range, achieving the lowest weight error (l). In summary, Weight-Init-L and Weight-Init-S reduce weight discrepancy through complementary mechanisms, outlier removal versus range compression, so their combination yields the lowest weight discrepancy.

### A.3 ACKNOWLEDGEMENT OF LLMs USAGE

We acknowledge the assistance of LLMs in polishing the paper writing and generating code used in our experiments.

