# OpenReview forum: "Compensate, Don't Reconstruct: Parameter- and Data-Efficient 2-bit LLM Quantization"
_ICLR.cc/2026/Conference — Submitted to ICLR 2026_

### Official Review · Reviewer_guY5 · 2025-10-27

**Soundness:** 2
**Presentation:** 4
**Contribution:** 3
**Rating:** 4
**Confidence:** 4

**Summary:**

This paper presents an in-depth empirical study of Quantization Error Compensation (QEC) and, based on the findings, introduces LG-QEC, an efficient and effective recipe for performing QEC in low-bit (e.g., 2-bit) large language models.

**Strengths:**

* This paper presents extensive and well-designed empirical studies on QEC methods, which offer valuable insights for researchers and practitioners in this field. In particular, the findings that hybrid approaches achieve impressive performance, and the analysis of how weight and activation discrepancies evolve with an increasing number of training tokens, are especially noteworthy.

* The proposed QEC recipe, LG-QEC, demonstrates strong empirical performance.

* The paper is very well-written and easy to follow.

**Weaknesses:**

* Some of the analyses behind the empirical results are not entirely persuasive to me.

The first concerns the concept of weight stability. The authors seem to regard weight stability as something essential for high-quality adaptation, mentioning that vector quantization performs better with adaptation than uniform quantization. However, I find this interpretation somewhat overstated. The superior results of vector quantization could simply stem from its inherent advantages over uniform quantization, rather than from better weight stability being a more suitable fit for QEC. The authors’ attempt to attribute vector quantization’s strength to weight stability does not sound convincing. In my view, to substantiate the unique role of weight stability in QEC, an experiment should compare two quantization methods where one shows poorer base quantization results but higher weight stability, and yet performs better under QEC than the other with the opposite characteristics.

Second, the rationale behind the design of LG-QEC is somewhat unclear.
The authors argue that end-to-end fine-tuning can be suboptimal because it must simultaneously optimize two objectives, weight and activation discrepancies, and therefore propose to decouple them. However, their actual approach, which first optimizes only activation discrepancy and then proceeds with end-to-end fine-tuning, does not appear to implement such decoupling. A true decomposition would be something like a first stage for weight discrepancy and a second stage for activation discrepancy. While I acknowledge that LG-QEC achieves strong empirical results, the justification and explanation of why it works appear somewhat speculative.

* The analysis and evaluations are performed exclusively on a single model, Llama-3-8B, leaving questions about generalizability to other architectures, particularly MoE models.

* Defining the budget solely in terms of parameter size reduces the paper’s practical relevance. To make the discussion more useful for practitioners, inference efficiency should also be considered. For instance, sparse adaptation methods may degrade inference throughput compared to LoRA even assuming the same parameter size, and vector quantization often leads to slower inference than scalar quantization. Incorporating such efficiency considerations would make the paper’s insights substantially more valuable.

**Questions:**

Directly related to the weaknesses I mentioned above.
I would like the authors to clarify whether weight stability is indeed an important factor for QEC, and how LG-QEC achieves a decoupling of the optimization processes for weight and activation discrepancies.
In addition, I would appreciate any discussion or speculation on how the presented findings might generalize to other model types, especially architectures beyond Llama-style dense transformers.

---

> ### Author Response · Authors · 2025-11-24
> **Response to the Reviewer's Comments**
>
> Thank you for your comments
>
> > Q1.  I would like the authors to clarify whether weight stability is indeed an important factor for QEC
>
> Lower weight stability leads to leads to data-dependent reconstruction and unstable adaptation dynamics, whereas higher weight stability enables stable activation alignment. Due to the large error inherent in 2-bit quantization, QAT reconstruction tends to relearn new functional representation when trained with a huge dataset, and this behavior is highly data-dependent. Figure 5(c) further shows that, under uniform quantization, QEC often induces activation drift in deeper layers to increase output similarity, which supports the view that reduced weight stability causes unstable adaptation. In contrast, vector quantization better preserves the original weights, resulting in a consistent decrease in activation discrepancy throughout training.
>
> > Q2. how LG-QEC achieves a decoupling of the optimization processes for weight and activation discrepancies.
>
> In LG-QEC, “decoupling” refers to separating weight fidelity and activation alignment during optimization, not to decoupling the two discrepancies themselves. In global optimization, QEC with a uniform quantizer often reduces the overall loss by improving activation alignment at the expense of weight fidelity. In contrast, vector quantizer, the enhanced quantization, mitigates this trade-off, and we aim to further enforce this effect through local optimization. Local optimization aligns activations within a limited scope, so it achieves better activation matching without substantially compromising weight fidelity. We agree with that the wording should be revised to make this distinction clearer.
>
> > Q3. would appreciate any discussion or speculation on how the presented findings might generalize to other model types, especially architectures beyond Llama-style dense transformers.
>
> We hypothesize that MoE does not exhibit a substantial difference from dense models because our method does not explicitly leverage architectural characteristics. However, as is well known, expert activations in MoE are often imbalanced. Therefore, for local optimization, it is important to use a calibration dataset that activates experts in a well-balanced manner to optimize every experts. LG-QEC’s adapter initialization is not activation-aware, so we do not expect a notable difference between dense and MoE models.
>
> > W4. Defining the budget solely in terms of parameter size reduces the paper’s practical relevance. To make the discussion more useful for practitioners, inference efficiency should also be considered. For instance, sparse adaptation methods may degrade inference throughput compared to LoRA even assuming the same parameter size, and vector quantization often leads to slower inference than scalar quantization. Incorporating such efficiency considerations would make the paper’s insights substantially more valuable.
>
> The table below reports inference cost results for Llama-3-8B. As vector quantizer itself is slower than the uniform quantizer, TTFT and TPOT are larger under vector quantization. Attaching adapters to a quantized model requires additional memory and computation, which increases the peak memory and latency. Also, in our implementation, LoRA and sparse adapter computations are performed sequentially rather than simultaneously, so this hybrid adapter incurs additional latency. Increasing the adapter size in the hybrid structure increases memory usage but does not significantly increase latency.
>
> **Inference cost (Model: Llama-3-8B)**
> |Quantizer|Adapter|Model Size(GB)|TTFT(ms)|TPOT(ms/token)|
> |:-:|:-:|:-:|:-:|:-:|
> |Uniform|-|2.8|496.70|314.31|
> |Uniform|LS(3.5%)|3.4|682.30|363.30|
> |Vector|-|2.8|798.01|559.59|
> |Vector|L(3.5%)|3.4|838.13|580.63|
> |Vector|LS(3.5%)|3.4|921.80|614.35|
> |Vector|LS(7.4%)|3.8|971.92|626.84|

---

> > ### Comment · Reviewer_guY5 · 2025-11-25
> >
> > Thank you for the detailed explanations.
> >
> > For Q1, I’m afraid my concerns regarding your explanation of weight stability are not fully resolved. It still feels as though the correlation is somewhat overstated and treated as if it implies causality. I reread Figure 5 and the associated explanations as you suggested, but I still find it difficult to identify empirical evidence demonstrating that weight stability is necessary for high-quality QEC.
> >
> > For Q2, thank you for the clarification. I hope this point will be clearly reflected in the revised version.
> > For Q3, thank you as well for the discussion; I agree with your points.
> >
> > For Q4, I am concerned that the overhead introduced by the adapters is more significant than I initially expected. Why is the memory overhead much larger than the parameter budget? And given the non-negligible memory and latency overhead (around 10%), how can you still argue for the practicality of the QEC approach itself?
> >
> > I would appreciate your additional comments. Thank you.

---

> > > ### Author Response · Authors · 2025-11-26
> > > **Response to the Reviewer's Second Comments**
> > >
> > > Thank you for your response. We agree with your concerns.
> > >
> > > > For Q1, I’m afraid my concerns regarding your explanation of weight stability are not fully resolved. It still feels as though the correlation is somewhat overstated and treated as if it implies causality. I reread Figure 5 and the associated explanations as you suggested, but I still find it difficult to identify empirical evidence demonstrating that weight stability is necessary for high-quality QEC.
> > >
> > > As we mentioned, QAT uses the original weights only as an initial point and does not explicitly maintain their behavior. When we conducted QAT and increased the number of training tokens, we obtained an average PPL of 18.6, but the average CSQA accuracy was only 50.35%. In contrast, when we increased the adapter size (which is not practical in real deployments but acceptable for analysis) and initialized it to reduce the weight discrepancy, we obtained an average PPL of 22.85 and an average CSQA accuracy of 54.79% without any training. This example indicates that QAT effectively relearns from the training dataset: even if PPL can be reduced by next-token training, a very large dataset is required to recover the original model’s behavior (e.g., CSQA accuracy). By contrast, keeping the original weights closer to their pre-quantization values through FP adapters helps better preserve the original behavior.
> > >
> > > > For Q4, I am concerned that the overhead introduced by the adapters is more significant than I initially expected. Why is the memory overhead much larger than the parameter budget? And given the non-negligible memory and latency overhead (around 10%), how can you still argue for the practicality of the QEC approach itself?
> > >
> > > We emphasize that QEC offers a practical trade-off: it can reach QAT-level accuracy with much smaller datasets, at the cost of some overhead. In our implementation, the trainable adapters use the BF16 data type, whereas the quantized weights are stored in 2 bits. That’s the reason of memory overhead is larger than trainable parameter ratio. We agree that the resulting adapter size and latency appear large, but we emphasize that prior PTQ methods struggle to recover accuracy at 2-bit precision, and QAT requires a very large scale dataset. QEC can achieve QAT-level accuracy with a much smaller dataset with some overhead. If we have sufficient high-quality data and computational resources to conduct large-scale training, QAT may indeed be a better option to achieve high accuracy; however, in scenarios where data or resources are are limited, QEC can be a practical alternative. Reducing this overhead is one of the key goals of our research, and further decreasing the required adapter size is an important direction for future work.

---

### Official Review · Reviewer_T5JN · 2025-10-29

**Soundness:** 3
**Presentation:** 2
**Contribution:** 2
**Rating:** 4
**Confidence:** 4

**Summary:**

This paper provides a detailed analysis of the Quantization Error Compensation (QEC) quantization method. The paper posits that traditional Quantization-Aware Training (QAT) performs "Reconstruction," whereas QEC performs "Compensation." Through observations from QEC experiments analyzing different adapter structures and initialization methods, the scalability of training parameters, and the selection of quantizers, the authors identify the key factors for improving the accuracy of the QEC quantization method. Finally, the authors propose LQ-QEC, employing a two-stage QEC approach to compensate for 2-bit LLM quantization, claiming to achieve high 2-bit quantization accuracy using minimal trainable parameters and a limited training token budget.

**Strengths:**

1.  **Novel Research Problem:** The focus is on the Quantization Error Compensation (QEC) problem. This strategy effectively addresses the issue of traditional Quantization-Aware Training (QAT) frameworks requiring substantial computational resources for extremely low-bit quantization, saving overhead and possessing practical value.
2.  **Rich Theoretical Analysis:** The paper provides a very detailed overview of different methods currently employed for Quantization Error Compensation (QEC). It analyzes various choices, such as different adapters (LoRA/Top-k), different initialization methods (L/S/LS), and different quantizers (uniform/vector). The experimental scope is extensive, yielding numerous valuable observations and insights.

**Weaknesses:**

1.  **Lacks Clear Writing Logic:** The core of the paper is the LQ-QEC quantization method (Section 5), but the exposition on LQ-QEC itself is relatively brief (e.g., how the budget is allocated between the first and second stages, more ablation study results, visualization of discrepancy values after adopting the two-stage scheme). Instead, the analysis in Section 4 takes up the majority of the content. Furthermore, within Section 4, Sections 4.1-4.2, which discuss the selection of different adapters and initialization methods, are not closely related to the core insights presented later. The authors should allocate more space to the core LQ-QEC method. Some introductory experiments and observations are overly lengthy, hindering in-depth reading.
2.  **Core Contribution is Unclear:** Logically, the paper seems to focus on improving the QEC method. However, the abstract or introduction emphasizes more that their method requires significantly fewer trainable parameters and training token budget compared to traditional QAT methods. This advantage should primarily be attributed to QEC itself, rather than being a specific contribution of this paper.
3.  **Lacks Deeper Investigation Across Bit-widths:** Although the paper's title is "2-bit LLM quantization," the insights observed in this paper, such as the increase in weight discrepancy and the decrease in activation during QEC training, are not analyzed across other bit-width scenarios.

**Questions:**

1.  **Regarding the Core Argument of the Paper:** The paper devotes significant space to discussing the trend of weight and activation discrepancy changes during the QEC fine-tuning process. Sections 4.3 and 4.4 visualize these trends, but the connection between the defined discrepancy values and the final end-to-end performance is not well established, making the argument less convincing. For example, in Figure 4(C), the magnitude of the increase in weight discrepancy is notably smaller than the decrease in activation discrepancy. What impact does this difference in magnitude have on the final performance? Furthermore, in Section 5, when proposing LQ-QEC, why is there no quantitative comparison of discrepancy changes to get related to the content presented in Section 4?
2.  **Regarding Experimental Setup:**

    (1) Why is there a lack of comparison with more training tokens for local optimization under the "Uniform quantizer" in Table 1? Why aren't experiments with local optimization conducted under the Uniform quantizer setting?

    (2) The number representing the QAT token consumption in the "ParetoQ" row of Table 2 is somewhat misleading. Since that setting and the paper's QEC method are entirely different (QEC requires a pretrained checkpoint as a base), directly listing "30B vs 16M tokens" could be slightly misleading.

    (3) Why is there no entry for the vector method *without* local optimization in this table? It seems the vector method alone is not proposed in this paper but originates from QuIP# (Tseng et al., 2024). (4) For Table 2, it is recommended to also provide downstream task metrics (e.g., CSQA and MMLU) as done in other tables.

---

> ### Author Response · Authors · 2025-11-24
> **Response to the Reviewer's Comments (1/2)**
>
> Thank you for your comments.
>
> > W1. Lacks Clear Writing Logic:
>
> We acknowledge that our two-stage optimization appears independent ot the choice of quantizer, adapter structure, or initialization method. However, in our analysis aimed at improving QEC, we investigated a range of adapter structure, initialization strategies, training budgets, and quantizer choices. Through these investigations, we compared weight discrepancies and activation discrepancies, and these analyses motivated the design of LG-QEC.
>
> > W2. Core Contribution is Unclear:
>
> We agree that parameter-efficiency and data-efficiency are not our contributions; rather, they are contributions of QEC. Nevertheless, there has been relatively little investigation of these aspects in the context of QAT and QEC.
>
> > W3. Lacks Deeper Investigation Across Bit-widths
>
> For 3- and 4-bit quantization, PTQ already achieves near-baseline accuracy. ParetoQ also indicates that 3- or 4-bit QAT primarily learns to compensate for quantization error, whereas lower-bit QAT (e.g., 2-bit) tends to exhibit a reconstruction-oriented learning pattern. Notably, even though 2-bit QAT can recover accuracy substantially when trained with a very large dataset, it still shows reconstruction behavior rather than compensation. Our QEC method recovers accuracy while preventing the weight discrepancy from increasing too much.
>
> > Q1. Regarding the Core Argument of the Paper:
>
> We would like to emphasize that Figure 4(c) shows that training with more tokens improves PPL and that activation discrepancy continues to decrease, while weight discrepancy increases. Of course, we agree that the magnitude of the increase in weight discrepancy is smaller than that of activation but these discrepancy. However, theses discrepancies vary depending on their dimensionality and statistics. For example, in Figure 5(c,d), the discrepancy of logits is much smaller than that of layer outputs because their dimensions and statistics are quite different. This is why we define both weight and activation discrepancies using relative L1 norm, so as to compare how much the weights and activations change relative to the initial baseline. The results indicate that increasing training tokens continuously decreases activation discrepancy but increases weight discrepancy.
>
> > Q2. (1) Why is there a lack of comparison with more training tokens for local optimization under the "Uniform quantizer" in Table 1? Why aren't experiments with local optimization conducted under the Uniform quantizer setting?
>
> We additionally tested local optimization with a uniform quantizer. For vector quantization, local optimization further improves accuracy after global optimization, especially on CSQA and MMLU. However, for uniform quantization, local optimization does not improve accuracy. We hypothesize that the quantization error of uniform quantization is too high, so local optimization distorts the adapters excessively, limiting their flexibility for global optimization. These results emphasize that, for QEC to reduce activation drifts, adopting a better quantizer (e.g. a vector quantizer) is necessary.
>
> **Updated Table 1 (Model: Llama-3-8B)**
> |Quantizer|Local Optim|Training Tokens|Wiki2 PPL(↓)|C4 PPL(↓)|ARC-C(↑)|ARC-E(↑)|HellaSwag(↑)|PIQA(↑)|WinoGrande(↑)|Avg. MMLU(↑)|
> |:-:|:-:|:-:|:-:|:-:|:-:|:-:|:-:|:-:|:-:|:-:|
> |Baseline|-|-|6.14|8.88|50.43|80.22|60.15|79.54|72.93|65.13|
> |Uniform|-|16M|11.15|13.92|34.64|68.14|49.86|74.48|66.30|40.17|
> |Uniform|-|32M|11.04|13.69|35.07|68.90|49.88|73.83|63.46|43.18|
> |Uniform|-|64M|11.16|13.53|35.67|68.64|50.76|75.41|65.35|41.23|
> |Uniform|o|16M|11.23|13.94|35.84|67.30|49.86|74.27|64.01|39.27|
> |Vector|-|16M|8.15|11.26|42.66|74.37|55.08|76.55|69.06|55.06|
> |Vector|-|32M|_8.11_|_11.15_|41.98|74.79|55.45|76.66|69.06|55.37|
> |Vector|-|64M|**8.08**|**11.04**|**43.43**|_75.59_|**55.72**|_76.99_|**69.53**|_55.63_|
> |Vector|o|16M|_8.11_|11.19|_43.09_|**75.63**|_55.52_|**77.20**|**69.53**|**55.65**|

---

> > ### Author Response · Authors · 2025-11-24
> > **Response to the Reviewer's Comments (2/2)**
> >
> > > Q2. (2) The number representing the QAT token consumption in the "ParetoQ" row of Table 2 is somewhat misleading. Since that setting and the paper's QEC method are entirely different (QEC requires a pretrained checkpoint as a base), directly listing "30B vs 16M tokens" could be slightly misleading
> >
> > ParetoQ states that they initialize QAT with pre-trained weights. Therefore, comparing our QEC with 16M tokens is not an unfair comparison.
> >
> > > Q2. (3) Why is there no entry for the vector method without local optimization in this table? It seems the vector method alone is not proposed in this paper but originates from QuIP# (Tseng et al., 2024).
> >
> > We acknowledge that our vector quantization is based on QuIP#. We did not added results without local optimization in Table 2 because we already compared effect of local optimization in Table 1.
> >
> > >  Q2. (4) For Table 2, it is recommended to also provide downstream task metrics (e.g., CSQA and MMLU) as done in other tables.
> >
> > This is the revised Table 2 with additional benchmarks and the baseline QEC results.
> >
> > **Table 2 with additional 2-bit QEC baselines (Model: Llama-3-8B)**
> >
> > |Quantization|Method|Training Token|Model Size(GB)|Wiki2 PPL(↓)|C4 PPL(↓)|ARC-C(↑)|ARC-E(↑)|HellaSwag(↑)|PIQA(↑)|WinoGrande(↑)|
> > |:-:|:-:|:-:|:-:|:-:|:-:|:-:|:-:|:-:|:-:|:-:|
> > ||Baseline|-|2.8|6.1|8.88|50.4|80.2|60.2|79.5|72.9|
> > |PTQ|RTN|-|2.8|2.25E+4|1.02E+4|20.7|26.5|25.9|53.5|49.7|
> > |PTQ|OmniQuant|-|2.8|61.79|52.91|19.3|36.1|32.9|59.0|51.9|
> > |PTQ|QuIP#|-|2.8|12.74|16.84|30.8|57.1|44.7|67.5|62.4|
> > |PTQ|QuaRot|-|2.8|14.95|27.77|24.6|57.9|36.8|65.2|60.6|
> > |PTQ|GPTQ|-|2.8|2.10E+2|4.10E+4|19.9|28.8|27.7|53.9|50.5|
> > |PTQ|AWQ|-|2.8|1.70E+6|2.10E+8|21.5|24.2|25.6|52.4|50.7|
> > |PTQ|Slim-LLM|-|2.8|39.7|110|26.1|35.4|28.9|57.1|56.6|
> > |PTQ|DB-LLM|-|2.8|13.6|19.2|28.2|59.1|42.1|68.9|60.4|
> > |PTQ|PB-LLM|-|2.8|24.7|79.2|17.2|37.8|29.8|57.0|52.5|
> > |QAT|EfficientQAT|16M|2.8|9.41|12.77|37.0|71.2|51.9|76.0|67.7|
> > |QEC|LoftQ|0.4M|3.4|18.01|20.63|28.0|58.7|42.8|69.7|58.8|
> > |QEC|LoftQ|16M|3.4|13.52|15.9|30.5|63.2|45.7|72.5|62.0|
> > |QEC|LoftQ|32M|3.4|13.23|15.32|31.1|65.2|46.6|72.4|61.8|
> > |QEC|PiSSA|16M|3.4|14.7|17.55|27.9|58.7|43.8|70.6|60.5|
> > |QEC|LG-QEC(3.5%)|16M|3.4|_8.46_|_11.49_|**43.4**|_74.8_|_54.8_|_76.6_|_68.8_|
> > |QEC|LG-QEC(7.4%)|16M|3.8|**8.11**|**11.19**|_43.1_|**75.6**|**55.5**|**77.2**|**69.5**|

---

### Official Review · Reviewer_EnF3 · 2025-11-03

**Soundness:** 2
**Presentation:** 3
**Contribution:** 2
**Rating:** 4
**Confidence:** 3

**Summary:**

This paper proposes LG-QEC, an efficient framework for 2-bit LLM quantization. The authors show that model performance depends more on activation discrepancy than on weight discrepancy, and design a compensation-based approach combining hybrid adapters (low-rank + sparse), vector quantization, and a local-global optimization scheme. Experiments on Llama-3-8B demonstrate that LG-QEC achieves ParetoQ-level performance (PPL ≈ 8.1 on WikiText-2) using only 16 M training tokens, drastically reducing both computation and parameter cost.

**Strengths:**

1. This paper has a clear motivation and identifies a genuine gap between QAT and QEC for 2-bit LLMs.
2. LG-QEC is conceptually clean. It combines hybrid adapters + vector quantization + local–global optimization and fit together nicely.
3. The method matches QAT performance with ~1/1800 of the training tokens. Ablations are thorough and convincing.

**Weaknesses:**

1. Experiments feel a bit narrow. Results are mainly WikiText-2, C4, and CSQA; I’d like to see broader coverage (reasoning, code, multilingual).
2. No hardware results. Everything is perplexity/accuracy, so the “on-device” efficiency claim isn’t fully demonstrated.

**Questions:**

1. The experiments mainly focus on WikiText-2, C4, and a few commonsense QA benchmarks. Have you tested LG-QEC on other domains (e.g., reasoning, coding, or multilingual datasets)? If not, do you expect the same activation-discrepancy insights to hold there?
2. Since the paper emphasizes on-device efficiency, could you provide empirical results on real hardware (e.g., inference latency, memory usage, or throughput) to substantiate the claimed deployment benefits?

---

> ### Author Response · Authors · 2025-11-24
> **Response to the Reviewer's Comments**
>
> Thank you for your comments.
>
> > W1. Experiments feel a bit narrow. Results are mainly WikiText-2, C4, and CSQA; I’d like to see broader coverage (reasoning, code, multilingual).
> Q1. The experiments mainly focus on WikiText-2, C4, and a few commonsense QA benchmarks. Have you tested LG-QEC on other domains (e.g., reasoning, coding, or multilingual datasets)? If not, do you expect the same activation-discrepancy insights to hold there?
>
> We did not conduct experiments on reasoning tasks. Since Llama-3-8B is not a dedicated reasoning model like DeepSeek-R1, performance may differ in such settings. Nevertheless, we hypothesize that for reasoning models with long chains of thought, activation errors may be less critical: even if an intermediate reasoning step is incorrect, the model may be able to detect the mistake and continue by generating an alternative reasoning path. Therefore, we expect that weight discrepancies could have a larger impact on reasoning models.
>
> > W2. No hardware results. Everything is perplexity/accuracy, so the “on-device” efficiency claim isn’t fully demonstrated.
> Q2. Since the paper emphasizes on-device efficiency, could you provide empirical results on real hardware (e.g., inference latency, memory usage, or throughput) to substantiate the claimed deployment benefits?
>
> The table below reports inference cost results for Llama-3-8B. As vector quantizer itself is slower than the uniform quantizer, TTFT and TPOT are larger under vector quantization. Attaching adapters to a quantized model requires additional memory and computation, which increases the peak memory and latency. Also, in our implementation, LoRA and sparse adapter computations are performed sequentially rather than simultaneously, so this hybrid adapter incurs additional latency. Increasing the adapter size in the hybrid structure increases memory usage but does not significantly increase latency.
>
> The table below reports inference cost results for Llama-3-8B. As vector quantizer itself is slower than the uniform quantizer, TTFT and TPOT are larger under vector quantization. Attaching adapters to a quantized model requires additional memory and computation, which increases the peak memory and latency. Also, in our implementation, LoRA and sparse adapter computations are performed sequentially rather than simultaneously, so this hybrid adapter incurs additional latency. Increasing the adapter size in the hybrid structure increases memory usage but does not significantly increase latency.
>
> **Inference cost (Model: Llama-3-8B)**
> |Quantizer|Adapter|Model Size(GB)|TTFT(ms)|TPOT(ms/token)|
> |:-:|:-:|:-:|:-:|:-:|
> |Uniform|-|2.8|496.70|314.31|
> |Uniform|LS(3.5%)|3.4|682.30|363.30|
> |Vector|-|2.8|798.01|559.59|
> |Vector|L(3.5%)|3.4|838.13|580.63|
> |Vector|LS(3.5%)|3.4|921.80|614.35|
> |Vector|LS(7.4%)|3.8|971.92|626.84|

---

### Official Review · Reviewer_SrtC · 2025-11-03

**Soundness:** 1
**Presentation:** 2
**Contribution:** 2
**Rating:** 4
**Confidence:** 4

**Summary:**

This paper propose LG-QEC that combines a hybrid adapter (low-rank and sparse matrix) and a local-global optimization strategy to align activations and suppress quantization errors. In 2-bit quantization. Experiments show LG-QEC achieves accuracy comparable to state-of-the-art QAT methods while using only a fraction of the training token budget and trainable parameters.

**Strengths:**

1. This paper is easy to follow
2. The insight "effectiveness is more dependent on minimizing activation than minimizing weight discrepancy alone" is well-motivated and validated through systematic experiments.
3. Both Uniform quantization and Vector quantization are applied in the experiments

**Weaknesses:**

1. **Inefficient and Potentially Misleading Sparsity**: The proposed method relies on unstructured sparsity within its hybrid adapters. This design choice is problematic for two reasons. First, _unstructured sparsity does not typically yield practical speedups or memory savings during training, as the computational graphs and gradient updates remain dense._ Second, the paper uses the number of elements (non-zero) as budget, which is an unrealistic metric for hardware efficiency. True memory benefits only materialize when sparsity exceeds a high threshold and supported by specialized hardware, which is not the case here. In fact, training two large, unstructured sparse matrices likely incurs a computational overhead exceeding that of standard QAT, as the dense representation of these sparse structures must be maintained and processed. This fundamentally undermines the paper's claims of training efficiency.
2. **Unfair Comparison and Overlooked Deployment Costs**: The experimental comparison is skewed. LG-QEC is evaluated while retaining its FP adapters (1GB in a 3.8GB model), whereas QAT methods are full quantized. A fair comparison requires merging the FP adapters back into the quantized weights and then re-quantizing the entire model to a true 2-bit state, ensuring all methods have identical computational footprints. The current setup masks potential "merge errors" and inflates the perceived performance of LG-QEC. The limited gains reported in Table 2 are questionable given the significant 1GB of FP parameters, raising concerns about the method's utility in a realistic, efficiency-focused deployment scenario.
3. **Lack of Efficiency Evaluation**: Given the substantial overhead introduced by the hybrid adapter structure (both sparse and low-rank), the claims of parameter and training efficiency are not substantiated by relevant metrics. The paper lacks critical experiments measuring actual training memory footprint, training latency, and—most importantly—inference latency and memory usage. Without benchmarking these practical hardware metrics against QAT and PTQ baselines, the proposed method's real-world viability remains unproven.
4. **Limited Model Scale and Task Diversity:** The evaluation is conducted primarily on Llama-3-8B. It is crucial to demonstrate the effectiveness of LG-QEC across a wider range of model scales, including both smaller (e.g., 1B, 3B) and, more importantly, larger models (e.g., 70B).
5. **Questionable Citation and Related Work**:  This paper states "... either low-rank or sparse adapters Li et al (a),; Guo et al; Zhang et al;". However, I read these references none of them do not use sparse adapters, are their correct?

**Questions:**

1. How do you expect LG-QEC to scale to much larger models (e.g., 70B)? Do you foresee new challenges, or will the same principles hold? Conversely, have you tested on smaller models (e.g., 1B-3B) where the parameter overhead of your adapters is relatively larger?
2. In the local optimization stage, you use only half the parameter budget. What is the intuition behind this?
3. The performance of your hybrid adapter seems highly dependent on the use of a high-quality vector quantizer. Does this mean LG-QEC is less effective or even fails with a standard uniform quantizer?

---

> ### Author Response · Authors · 2025-11-26
> **Response to the Reviewer's Comments (1/2)**
>
> Thank you for your comments.
>
> > W1. Inefficient and Potentially Misleading Sparsity:
>
> First, our implementation of the sparse adapter is based on SHiRA. We acknowledge that we did not implement a specialized kernel for sparse operation with a quantized model. We agree that sparse training computes dense gradients during backpropagation. However, the dominant memory overhead in training comes from optimizer states, which are proportional to the number of trainable parameters. In our sparse training setting, even though the gradients are computed densely, optimizer states are added only for the sparse trainable parameters. Therefore, sparse training requires substantially less memory than full qat.
> In addition, similar to Weight-Init-S and Error-Init-LS, we initialize 2 sparse adapters, but we are not training these 2 sparse adapters independently. We can easily merge them into a single sparse adapter. Thus, in practice, we utilize only a single sparse adapter for both training and inference.
>
> > W2. Unfair Comparison and Overlooked Deployment Costs:
>
> We do not merge adapters into quantized models, so there is no “merge error”. QEC uses a small FP adapter; therefore, it inevitably increases model size and latency. Nevertheless, it can improve accuracy without large-scaling dataset training, unlike QAT methods (e.g. ParetoQ). For extremely low-bit quantized models, using FP adapters while acknowledging this overhead is already adopted in industry (e.g. Apple Foundation Model’s Accuracy Recovery Adapters). We agree that the adapter size in our results is relatively large, so we added results with smaller adapters. The results show that our method remains competitive even with smaller adapter sizes.
>
> **LG-QEC with different adapter size  (Model: Llama-3-8B)**
> |Method|Model size|Avg. PPL(↓)|Avg. CSQA(↑)|
> |:-:|:-:|:-:|:-:|
> |Baseline|16GB|7.51|68.65|
> |LG-QEC(3.5%)|3.4GB|9.98|63.67|
> |LG-QEC(7.4%)|3.8GB|9.65|64.19|
>
> > W3. Lack of Efficiency Evaluation:
>
> The table below presents training cost results for Llama-3-8B. Because QAT trains all weight parameters, its required memory is much larger than that of QEC methods. Since inference with vector quantization is slower than with uniform quantization, QEC with a vector quantizer also requires more time. Nevertheless, because vector quantization yields higher accuracy than uniform quantization, we choose to use vector quantization in our method.
>
> **Training cost (Model: Llama-3-8B)**
> |Training|Memory|16M Token Training Time|
> |:-:|:-:|:-:|
> |QAT|59.68 GB|190m|
> |Uniform+LS(3.5%)|21.12 GB|180m|
> |Vector+LS(3.5%)|21.12 GB|253m|
> |Vector+LS(7.4%)|23.94 GB|260m|
>
> The table below reports inference cost results for Llama-3-8B. As vector quantizer itself is slower than the uniform quantizer, TTFT and TPOT are larger under vector quantization. Attaching adapters to a quantized model requires additional memory and computation, which increases the peak memory and latency. Also, in our implementation, LoRA and sparse adapter computations are performed sequentially rather than simultaneously, so this hybrid adapter incurs additional latency. Increasing the adapter size in the hybrid structure increases memory usage but does not significantly increase latency.
>
> **Inference cost (Model: Llama-3-8B)**
> |Quantizer|Adapter|Model Size(GB)|TTFT(ms)|TPOT(ms/token)|
> |:-:|:-:|:-:|:-:|:-:|
> |Uniform|-|2.8|496.70|314.31|
> |Uniform|LS(3.5%)|3.4|682.30|363.30|
> |Vector|-|2.8|798.01|559.59|
> |Vector|L(3.5%)|3.4|838.13|580.63|
> |Vector|LS(3.5%)|3.4|921.80|614.35|
> |Vector|LS(7.4%)|3.8|971.92|626.84|

---

> ### Author Response · Authors · 2025-11-26
> **Response to the Reviewer's Comments (2/2)**
>
> > W4. Limited Model Scale and Task Diversity:
>
> When comparing LG-QEC on Llama-3.2-3B, QAT still failed to recover accuracy with a small dataset. In contrast, QEC improved accuracy using only a small dataset, and local optimization further increased accuracy.
>
> **QAT and QEC results (Model: Llama-3.2-3B)**
> |Method|Training Token|Avg. PPL(↓)|Avg. CSQA(↑)|
> |:-:|:-:|:-:|:-:|
> |Baseline|-|9.13|63.68|
> |QAT|16M|24.28|44.51|
> |QEC|16M|11.95|58.7|
> |LG-QEC|16M|**11.91**|**58.96**|
>
>
> > W5. Questionable Citation and Related Work:
>
> We agree that the previous explanation of related work did not match well. We revise it as follows:
> LoRA is commonly used in prior QEC methods (e.g. LoftQ, LQ-LoRA, and LQER), whereas sparse adapters have not been thoroughly explored. RoSA employed both LoRA and a sparse adapter, but did not consider quantization error when initializing adapters.
>
> > Q1. In the local optimization stage, you use only half the parameter budget. What is the intuition behind this?
>
> When we set only half of the parameter budget as trainable for local optimization, we observed better performance than when using all parameters as trainable. The table below shows this comparison. We hypothesize that when all adapter parameters are trainable during local optimization, minimizing the MSE loss can introduce excessive distortion, which reduces flexibility for subsequent global optimization. In contrast, using only half of adapter parameters as trainable freezes the other half (both trainable and frozen parameters are initialized to reduce weight discrepancy). These frozen weights act as a regularizer; preventing excessive distortion while still reducing the MSE loss of the layer output. As a result, this setting yields better accuracy after global optimization
>
> **Local optimization - trainable parameter ablation (Model: Llama-3-8B)**
> |Local Optimization - Trainable|Avg. PPL(↓)|Avg. CSQA(↑)|
> |:-:|:-:|:-:|
> |Half|9.98|63.67|
> |All|9.99|63.54|
>
> > Q2. The performance of your hybrid adapter seems highly dependent on the use of a high-quality vector quantizer. Does this mean LG-QEC is less effective or even fails with a standard uniform quantizer?
>
> We agree that the accuracy improvement from using a vector quantizer is substantial. However, as discussed in Section 4.4, we found that global optimization with a more precise quantizer, such as a vector quantizer, helps prevent the activation drift observed with a uniform quantizer and enables further improvements.
> In our experiments, uniform quantization corresponds to simple RTN without further optimization. If we apply more advanced uniform quantization methods such as QAT or OmniQuant, accuracy with a uniform quantizer could be improved as well.

---

> > ### Comment · Reviewer_SrtC · 2025-11-26
> > **Official Comments by Reviewer**
> >
> > Thank the authors for posting their response. I read the revised paper and the rebuttal. However, several of my key concerns remain unaddressed.
> >
> > 1. The authors state that memory is "substantially less" because "optimizer states are added only for the sparse trainable parameters." This claim needs further clarification regarding the exact source of the memory savings.
> >
> > 2. The authors clarify that "We do not merge adapters into quantized models," and show that the sparse adapter does not significantly increase latency (compared to non-adapter in the third table ) and reduces memory usage (compared to FP model in the first table).  However, if the sparse adapters are not merged, inference requires executing both the quantized weight and one/two unstructured FP sparse matrices that have the same shape as the weight matrix. As I know (if I am wrong, please correct me ), this kind of unstructured sparsity is not well-supported by GPU hardware, and the SpMM (Sparse Matrix-Matrix Multiplication) operations would be very inefficient. So, how does the unmerged model achieve memory saving compared to the FP model in the first table, and does not introduce significant latency overhead in the third table?

---

> > > ### Author Response · Authors · 2025-11-28
> > > **Response to the Reviewer's Second Comments**
> > >
> > > Thank you very much for your constructive comments and for the opportunity to clarify our method.
> > >
> > > > 1. The authors state that memory is "substantially less" because "optimizer states are added only for the sparse trainable parameters." This claim needs further clarification regarding the exact source of the memory savings.
> > >
> > > For fine-tuning a weight matrix, the required training memory consists of the weights, gradients, and optimizer states, all of which are proportional to the number of parameters in the matrix. For example, if we use the BF16 data type, each parameter requires BF16 weight, BF16 gradient, and 2xBF16 optimizer states. Since BF16 is 2 bytes, this corresponds to 2 (weight) + 2 (gradient) + 4 (optimizer states) = 8 bytes per parameter. Thus, if the matrix has 100 parameters, the required memory for training is approximately 800 bytes: (2+2+4)x100.
> > >
> > > If we conduct sparse training with sparsity 2 (i.e., only 2 of the 100 parameters are trainable), we still need to store the full weights and gradients, but we only maintain optimizer states for the 2 trainable parameters. As you mentioned, the full gradient must be computed, but only 2 parameters are actually updated. In this case, the required memory is approximately 408 bytes: (2+2)x100+4x2. Here, (2+2) x 100 accounts for the full BF16 weights and gradients for all 100 parameters, and 4x2 accounts for the optimizer states of the 2 trainable parameters.
> > >
> > > For QEC, we quantize and freeze the base weights. The required memory for the 2-bit quantized weights of 100 parameters is 25 bytes (2x100 bits = 200 bits = 25 bytes). The sparse adapter uses BF16 weights for the 2 trainable parameters, which requires 2 x 16 bits = 4 bytes. In addition, we still need gradients for all parameters (2x100=200 bytes) and optimizer states for the 2 trainable parameters (4x2=8bytes). Therefore, the total required memory for QEC is (25 + 4 + 2 x 100 + 4 x 2) = 237 bytes. Of course the exact memory comparison will depend on the sparsity level. However, this example illustrates our main point: even though sparse training requires computing dense gradients, the training-time memory usage is still substantially lower than that of full QAT.
> > >
> > >
> > > > 2. The authors clarify that "We do not merge adapters into quantized models," and show that the sparse adapter does not significantly increase latency (compared to non-adapter in the third table ) and reduces memory usage (compared to FP model in the first table). However, if the sparse adapters are not merged, inference requires executing both the quantized weight and one/two unstructured FP sparse matrices that have the same shape as the weight matrix. As I know (if I am wrong, please correct me ), this kind of unstructured sparsity is not well-supported by GPU hardware, and the SpMM (Sparse Matrix-Matrix Multiplication) operations would be very inefficient. So, how does the unmerged model achieve memory saving compared to the FP model in the first table, and does not introduce significant latency overhead in the third table?
> > >
> > > First, the memory savings primarily come from sparsity. We only introduce additional sparse adapters, and their memory requirement is low because they contain very few trainable parameters. At the same time, by quantizing the base weights from BF16 to INT2, the memory compression achieved on the base model is much larger than the overhead introduced by the sparse adapters.
> > >
> > > Regarding latency, we acknowledge that using adapters introduces an additional computation path. Nevertheless, using FP adapters (typically LoRA) on top of quantized models is already widely adopted in both research and industry, due to their strong performance in improving accuracy and task adaptation. For sparse computation, the additional latency overhead relative to LoRA is not critical, thanks to the high sparsity of the adapter. However, if we increase the size of the sparse adapter, the latency overhead could be significant. In our comparison between "LoRA only" and "LoRA + Sparse Adapter" in the table, the additional latency overhead of the sparse adapters is not critical. Developing more efficient sparse kernels and implementations is an important direction that we consider as future work.

---

> > > > ### Comment · Reviewer_SrtC · 2025-11-28
> > > > **Response to Authors**
> > > >
> > > > > If we conduct sparse training with sparsity 2 (i.e., only 2 of the 100 parameters are trainable), we still need to store the full weights and gradients, but we only maintain optimizer states for the 2 trainable parameters. As you mentioned, the full gradient must be computed, but only 2 parameters are actually updated. In this case, the required memory is approximately 408 bytes: (2+2)x100+4x2. Here, (2+2) x 100 accounts for the full BF16 weights and gradients for all 100 parameters, and 4x2 accounts for the optimizer states of the 2 trainable parameters.
> > > >
> > > > This calculation is highly idealistic and impractical on real hardware and ML frameworks like GPUs and PyTorch. As I know, modern systems are built for vectorization and parallel operations. It's inefficient and often impossible to maintain a sparse set of trainable parameters within a matrix to train solely on them, without computing and saving optimizer states for the other elements.
> > > >
> > > > Moreover, the calculation completely ignores the memory required for the index that identifies the trainable parameters. For a sparsity of 2, the index overhead might be negligible, but for realistic, large-scale sparsity, the storage for these data structures can be substantial and must be included in any accurate memory analysis.
> > > >
> > > > Therefore, I'm curious where this memory saving and slight latency overhead actually come from.

---

> > > > > ### Author Response · Authors · 2025-12-01
> > > > > **Response to the Reviewer's Third Comments**
> > > > >
> > > > > Thank you for your comments.
> > > > >
> > > > > We would like to clarify that we utilize PyTorch's built-in sparse features to handle sparse parameter storage. Specifically, we store only the values of the sparse parameters along with their indices, which allows us to avoid allocating memory for the full dense matrix. For example, in the case of 2% sparsity within a 100-parameter matrix, only 2 weight values and 2 indices are stored. As each trainable parameter is handled independently by the optimizer, optimizer states are maintained only for parameters that are actually updated. Therefore, in sparse training, we can avoid allocating memory of optimizer states for the non-trainable full matrix.
> > > > >
> > > > > In our QEC approach, the sparse adapter is small relative to the full model. For instance in a configuration where only 3.5% of the base weight is trainable, the sparsity of the sparse adapter itself is 2%. While storing additional indices does introduce a minor memory overhead, the high sparsity makes this overhead negligible in practice. We recognize that further reducing the storage cost of index information is a meaningful area for future improvement.

---

### Official Review · Reviewer_GXWA · 2025-11-05

**Soundness:** 3
**Presentation:** 3
**Contribution:** 2
**Rating:** 4
**Confidence:** 4

**Summary:**

This paper argues the "reconstruction" paradigm of Quantization-Aware Training (QAT) for 2-bit LLM quantization, advocating for an efficient "compensation" approach (QEC) . The paper proposes: minimizing activation discrepancy is a more reliable predictor of final accuracy than minimizing weight discrepancy alone. Based on this, the paper proposes LG-QEC, a framework that decouples these objectives. It first uses a refined vector quantizer to suppress initial weight errors. Then, it employs a two-stage local-global optimization strategy to efficiently align activation distributions. The LG-QEC achieves accuracy comparable to state-of-the-art QAT while using 1875x less training data and a fraction of the trainable parameters.

**Strengths:**

1. The paper's performance in llama3-8B is superior in data and parameter efficiency.
2. The paper has clear presenation and writing, good motivation.
3. Rigorous Empirical Analysis: The paper is grounded in a rigorous empirical study (Sec. 4) . It methodically investigates the interplay between adapter architecture , initialization strategies , and quantizer design, building a foundation for the method.
4. Effective Decoupled Framework: The LG-QEC framework offers an effective solution by "decoupling" the quantization problem . It uses a vector quantizer to suppress weight discrepancies and a two-stage (local-global) optimization to align activations, providing a stable and efficient pathway to high accuracy .

**Weaknesses:**

1. Limited Empirical Validation: The framework's generalizability is unsubstantiated, as validation is confined to a single model (Llama-3-8B).
2. Critical Dependency on Vector Quantization: The method's success is critically contingent on a sophisticated vector quantizer (VQ). Performance collapses with a standard uniform quantizer (PPL > 11 vs. 8.1) , indicating the framework is only effective after VQ has suppressed initial weight errors.
3. Methodological Complexity: The complete LG-QEC pipeline—integrating a specialized VQ, a hybrid adapter, and a two-stage optimization procedure—is significantly more complex than standard QAT, posing a barrier to practical implementation.
4. For low-bit quantization of large language models, employing vector quantization appears to be a straightforward approach. Looks like this method is the combination of vector quantization and QEC-like method (e.g., LoftQ).

**Questions:**

1. What's the LG-QEC performance on other different scale models, such as llama3-1B, 3B, 70B. and multi zero-shot tasks (like lm-evaluation-harness).
2. What is the meaning of "LG" in LG-QEC?
3. Marginal Efficacy of L-G Optimization: The core two-stage (L-G) optimization strategy yields negligible empirical gains. Data shows this complex procedure only improved C4 PPL from 8.15 to 8.11, suggesting the performance benefits stem almost entirely from the VQ choice, not the L-G strategy?
4. Increased Inference Memory Footprint: The reliance on adapters results in a larger final model size (3.8GB) compared to the QAT baseline (2.8GB), conflicting with the primary goal of edge deployment.
5. Large language models are sensitive to activation quantization. Therefore, the first insight—that optimizing the discrepancy of activation values would be effective—does not appear to yield meaningful insights. Moreover, directly correlating output-side activation discrepancies has already been discussed in some prior quantization works (e.g., PD-Quant, CVPR 2023, LiDAR-PTQ, ICLR 2024). The authors should address this discrepancy in their discussion.
6. what is the training/finetuing time for LG-QEC?
7. Looks like the LG-QEC is weight-only quantization, what is the final memory saving ratio?

---

> ### Author Response · Authors · 2025-11-26
> **Response to the Reviewer's Comments (1/2)**
>
> Thank you for your comments.
>
> > W2. Critical Dependency on Vector Quantization: The method's success is critically contingent on a sophisticated vector quantizer (VQ). Performance collapses with a standard uniform quantizer (PPL > 11 vs. 8.1) , indicating the framework is only effective after VQ has suppressed initial weight errors.
> W4. For low-bit quantization of large language models, employing vector quantization appears to be a straightforward approach. Looks like this method is the combination of vector quantization and QEC-like method (e.g., LoftQ).
>
> We agree that the accuracy improvement from using a vector quantizer is substantial. Section 4.4, however, we found that global optimization with a more precise quantizer, such as a vector quantizer, helps prevent the activation drift observed with a uniform quantizer and enables further improvements.
>
>
> > W3. Methodological Complexity: The complete LG-QEC pipeline—integrating a specialized VQ, a hybrid adapter, and a two-stage optimization procedure—is significantly more complex than standard QAT, posing a barrier to practical implementation.
>
> We acknowledge that LG-QEC is more complex than QAT. However, recovering accuracy with QAT typically requires very large datasets (e.g., ParetoQ), whereas our method achieves comparable accuracy with far fewer training tokens and lower GPU memory requirements. The table below reports the required time for QAT and QEC for Llama-3-8B. In our current local-optimization implementation, each decoder layer is optimized sequentially; this step can be parallelized, which would further reduce the runtime.
>
> **Required time and accuracy (Model: Llama-3-8B)**
> |Method|Local(4M token)|Global(16M token)|Avg. CSQA(↑)|
> |:-:|:-:|:-:|:-:|
> |QAT|-|190m|47.41|
> |QEC|-|260m|63.54|
> |LG-QEC|213m|260m|64.19|
>
> > Q1. What's the LG-QEC performance on other different scale models, such as llama3-1B, 3B, 70B. and multi zero-shot tasks (like lm-evaluation-harness).
>
> **QAT and QEC results (Model: Llama-3.2-3B)**
> |Method|Training Token|Avg. PPL(↓)|Avg. CSQA(↑)|
> |:-:|:-:|:-:|:-:|
> |Baseline|-|9.13|63.68|
> |QAT|16M|24.28|44.51|
> |QEC|16M|11.95|58.7|
> |LG-QEC|16M|**11.91**|**58.96**|
>
> > Q2. What is the meaning of "LG" in LG-QEC?
>
> Here, “LG” in LG-QEC stands for Local and Global.
>
>
> > Q3. Marginal Efficacy of L-G Optimization: The core two-stage (L-G) optimization strategy yields negligible empirical gains. Data shows this complex procedure only improved C4 PPL from 8.15 to 8.11, suggesting the performance benefits stem almost entirely from the VQ choice, not the L-G strategy?
>
> We agree that the PPL improvement of local optimization is not large. As local optimization does not directly learn the next token prediction loss, PPL, which measures how well the model predicts the next token, can remain similar with or without local optimization, because both settings learn the same next-token objective during global optimization. Nevertheless, local optimization makes the output of each layer in the quantized model closer to the baseline model’s behavior, which leads to improved downstream accuracy such as CSQA or MMLU.
>
>
> > Q4. Increased Inference Memory Footprint: The reliance on adapters results in a larger final model size (3.8GB) compared to the QAT baseline (2.8GB), conflicting with the primary goal of edge deployment.
>
> We acknowledge that the adapter size used in our experiment is relatively large.Therefore, we added results using a 3.4% trainable-parameter setting for Llama-3-8B in the table below. Our method remains effective with a smaller adapter.
>
> **LG-QEC with different adapter size  (Model: Llama-3-8B)**
> |Method|Model size|Avg. PPL(↓)|Avg. CSQA(↑)|
> |:-:|:-:|:-:|:-:|
> |Baseline|16GB|7.51|68.65|
> |LG-QEC(3.5%)|3.4GB|9.98|63.67|
> |LG-QEC(7.4%)|3.8GB|9.65|64.19|

---

> ### Author Response · Authors · 2025-11-26
> **Response to the Reviewer's Comments (2/2)**
>
> > Q5. Large language models are sensitive to activation quantization. Therefore, the first insight—that optimizing the discrepancy of activation values would be effective—does not appear to yield meaningful insights. Moreover, directly correlating output-side activation discrepancies has already been discussed in some prior quantization works (e.g., PD-Quant, CVPR 2023, LiDAR-PTQ, ICLR 2024). The authors should address this discrepancy in their discussion.
>
> In our experiments, we do not quantize activations. We agree that reducing activation discrepancy as an optimization objective has already been explored in several prior methods (e.g. AdaRound[1], ApiQ[2], etc.). However, our goal was to investigate the relationship between weight discrepancy, activation discrepancy, and final performance under different adapter initialization. We found that Weight-Init-L reduces weight discrepancy more than other methods, but does not improve activation discrepancy; consequently, it yields poorer final accuracy than other approaches. Therefore, we conclude that reducing weight discrepancy does not necessarily guarantee a decrease in activation discrepancy.
> In addition, through our analysis of quantizer choice, we observed that a vector quantizer, showing smaller weight and activation discrepancies than a uniform quantizer, exhibits less activation drift after training. This suggests that training with a less precise quantizer can cause activation drift from the baseline model, which may reduce final accuracy. These are the findings we would like to emphasize.
>
> References
>
> [1] Up or Down? Adaptive Rounding for Post-Training Quantization, Nagel, Markus, et al., ICML 2020
>
> [2] ApiQ: Finetuning of 2-Bit Quantized Large Language Model, Liao, Baohao, et al., EMNLP 2024
>
> > Q6. what is the training/finetuing time for LG-QEC?
>
> The next table reports the training time for QEC for Llama-3-8B. For global optimization, we used 8192 samples with a sequence length of 2048 and a batch size of 1. Inference with vector quantization is slower than with uniform quantization, so training with vector quantizer takes longer. Nevertheless, because vector quantization yields higher accuracy than uniform quantization, we choose to use vector quantization in our method.
>
> **Training cost (Model: Llama-3-8B)**
> |Training|16M Token Training Time|
> |:-:|:-:|
> |QAT|190m|
> |Uniform+LS(3.5%)|180m|
> |Vector+LS(3.5%)|253m|
> |Vector+LS(7.4%)|260m|

---

### Official Review · Reviewer_8Yj5 · 2025-11-06

**Soundness:** 3
**Presentation:** 3
**Contribution:** 2
**Rating:** 2
**Confidence:** 4

**Summary:**

The paper proposes LG-QEC, a framework for efficient 2-bit quantization of LLMs using Quantization Error Compensation. It achieves perplexity of 8.1 on WikiText-2 with only 16M training tokens—a 1,875× reduction compared to  state-of-the-art methods requiring 30B tokens. The key insight is that minimizing activation discrepancy (rather than weight discrepancy) is crucial for effective quantization error compensation.

**Strengths:**

Achieving comparable performance to methods using 30B tokens with only 16M tokens represents major data efficiency gains.

The hybrid adapter structure and two-stage local-global optimization are clearly justified through ablations, showing complementary benefits and principled division of responsibilities.

**Weaknesses:**

The paper provides rigorous analysis showing that activation discrepancy correlates more strongly with final performance than weight discrepancy. However, this obervervation has already been found by many previous papers. It is far away from a new finding.

Only compares against RILQ from the QEC family, missing critical recent methods like PiSSA, LQ-LoRA, and LQER. No head-to-head comparisons under identical settings.

**Questions:**

refer to weaknesses

---

> ### Author Response · Authors · 2025-11-24
> **Response to the Reviewer's Comments**
>
> Thank you for your comments.
>
> > Weakness: The paper provides rigorous analysis showing that activation discrepancy correlates more strongly with final performance than weight discrepancy. However, this obervervation has already been found by many previous papers. It is far away from a new finding.
>
> As you pointed out, reducing activation (linear output) discrepancy is an optimization objective in several prior methods (e.g., AdaRound [1], GPTQ, ApiQ [2], etc.), and this objective often improves final performance. What we aimed to investigate is the relationship between weight discrepancy, activation discrepancy, and final performance under different adapter initializations. We found that PiSSA(Weight-Init-L) reduces weight discrepancy more than other methods, but does not improve activation discrepancy; consequently, it yields poorer final accuracy than other approaches. Therefore, we conclude that reducing weight discrepancy does not necessarily guarantee a decrease in activation discrepancy.
> In addition, through our analysis of quantizer choice, we observed that a vector quantizer, showing smaller weight and activation discrepancies than a uniform quantizer, exhibits less activation drift after training. This suggests that training with a less precise quantizer can cause activation drift from the baseline model, which may reduce final accuracy. Theses are the findings we would like to emphasize.
>
> References
>
> [1] Up or Down? Adaptive Rounding for Post-Training Quantization, Nagel, Markus, et al., ICML 2020
>
> [2] ApiQ: Finetuning of 2-Bit Quantized Large Language Model, Liao, Baohao, et al., EMNLP 2024
>
>
> > Weakness: Only compares against RILQ from the QEC family, missing critical recent methods like PiSSA, LQ-LoRA, and LQER. No head-to-head comparisons under identical settings.
>
> We compared multiple adapter initialization methods in section 4.2. LoftQ and LQ-LoRA both initialize LoRA by decomposing the quantization error matrix, which corresponds to  our Error-Init-L setting. In contrast, PiSSA performs SVD on the original weight, initializes LoRA using high singular-value vectors, and then quantizes the remaining weight; this matches our Weight-Init-L setting. Figure 3 shows that PiSSA fails to reduce activation discrepancy and leads to worse PPL results.  We have added PiSSA trained with 16M tokens to Table 2. In our implementation, PiSSA’s Weight-Init-L performs poorly  at 2-bit quantization, which is consistent with the results reported in LowRA[1].
>
> **Table 2 with additional 2-bit QEC baselines (Model: Llama-3-8B)**
> |Quantization|Method|Training Token|Model Size(GB)|Wiki2 PPL(↓)|C4 PPL(↓)|ARC-C(↑)|ARC-E(↑)|HellaSwag(↑)|PIQA(↑)|WinoGrande(↑)|
> |:-:|:-:|:-:|:-:|:-:|:-:|:-:|:-:|:-:|:-:|:-:|
> ||Baseline|-|2.8|6.1|8.88|50.4|80.2|60.2|79.5|72.9|
> |QAT|EfficientQAT|16M|2.8|9.41|12.77|37.0|71.2|51.9|76.0|67.7|
> |QEC|LoftQ|0.4M|3.4|18.01|20.63|28.0|58.7|42.8|69.7|58.8|
> |QEC|LoftQ|16M|3.4|13.52|15.9|30.5|63.2|45.7|72.5|62.0|
> |QEC|LoftQ|32M|3.4|13.23|15.32|31.1|65.2|46.6|72.4|61.8|
> |QEC|PiSSA|16M|3.4|14.7|17.55|27.9|58.7|43.8|70.6|60.5|
> |QEC|LG-QEC(3.5%)|16M|3.4|_8.46_|_11.49_|**43.4**|_74.8_|_54.8_|_76.6_|_68.8_|
> |QEC|LG-QEC(7.4%)|16M|3.8|**8.11**|**11.19**|_43.1_|**75.6**|**55.5**|**77.2**|**69.5**|
>
> References
>
> [1] LowRA: Accurate and Efficient LoRA Fine-Tuning of LLMs under 2 Bits, Zhou, Zikai, et al., ICML 2025

---

### Comment · Area_Chair_opmJ · 2025-11-26
**Please take a moment to read the authors’ responses**

Dear Reviewers,

I hope this message finds you well. As the Area Chair for this submission, I would like to kindly remind you that the author rebuttal is now available. Please take a moment to read the authors’ responses and, if necessary, update your reviews accordingly.

Thank you very much for your time and for contributing to the quality of ICLR 2026.

Best regards,

Your AC

---

### Comment · Area_Chair_opmJ · 2025-11-28

Dear authors and reviewers,

Please remain professional and refrain from being influenced by the event. If anyone violates the rules, please let me know, and I will flag and report it to the Program Chairs.

Your AC

---

### Meta-Review · Area_Chair_N4AD · 2026-01-04

**Summary:**

The reviewers’ concerns that informed the suggested decision can be summarized as follows:

1. Multiple reviewers noted that the key observation, activation discrepancy being more predictive of performance than weight discrepancy, has been discussed in several prior quantization works. The paper’s contribution was therefore viewed as incremental rather than conceptually new.

2. The method’s effectiveness relies critically on a sophisticated vector quantizer. Performance degrades substantially with a standard uniform quantizer, raising concerns that LG-QEC’s gains stem primarily from VQ rather than the proposed local–global (LG) optimization framework.

3. Reviewers raised concerns that the use of FP adapters (up to ~1GB) undermines claims of memory and inference efficiency, particularly when compared to fully quantized QAT baselines. Unstructured sparsity was also viewed as unlikely to yield real hardware benefits.

4. The empirical validation is concentrated mainly on LLaMA-3-8B, with limited evidence that the method generalizes to larger (e.g., 70B) or smaller models, or to a broader set of downstream tasks.

5. The full LG-QEC pipeline (vector quantization + hybrid adapters + local-global optimization) is substantially more complex than existing approaches, while the additional gains from the LG optimization stage itself are relatively small.

**Reviewer Concerns:**

Concerns addressed by the rebuttal:

1. The authors added comparisons and clarified that several prior methods (e.g., PiSSA, LQ-LoRA) correspond to specific initialization settings already evaluated. Additional results were provided, partially addressing fairness concerns.

2. Training and inference memory usage, latency (TTFT/TPOT), and training time were added, improving transparency and addressing requests for concrete efficiency metrics.

3. The authors explained why local optimization yields limited PPL improvement but can improve downstream task accuracy, addressing confusion about its marginal numerical impact.

Concerns still outstanding:

1. Despite clarifications, the core insight remains close to existing literature, and the rebuttal does not convincingly establish a fundamentally new principle beyond prior activation-alignment–based quantization methods.

2. The method retains sizable FP adapters at inference, while QAT baselines are fully quantized. Even with added discussion, this weakens claims of practical efficiency and edge deployment suitability.

3. The rebuttal confirms, rather than alleviates, that LG-QEC’s success depends heavily on VQ. This raises concerns that the proposed framework is not robust across quantizer choices.

4. Evidence for effectiveness on larger-scale models and more diverse tasks remains limited, and key scalability questions remain unanswered.

**Reviewer Scores:**

Based on the reviews and rebuttal, the following score adjustments are likely if reviewers had fully participated in discussion:

 Reviewer 8Yj5 (initial rating: 2  reject):
Likely increase to 4, acknowledging added baselines and clarifications, but still below acceptance due to novelty concerns.

Reviewer GXWA (initial rating: 4  marginally below threshold):
Likely remain at 4, or at most increase slightly, as core concerns about VQ dependency, complexity, and limited gains persist.

Reviewer SrtC (initial rating: 4  marginally below threshold):
Likely remain at 4, given continued concerns about unstructured sparsity, unfair efficiency comparisons, and limited deployment relevance.

No reviewer is likely to move clearly above the acceptance threshold, and the overall score distribution would remain below the bar.

---

### Decision · Program_Chairs · 2026-01-26

Reject